# Cyclic Oxidation Behavior of Conventional and Niobium-Modified MAR-M246 Superalloy at 900 and 1000 °C

Filipe Augusto de Faria Cunha [1,*], Rodrigo de Andrade Reis [1,2], Samantha Pinto Gonçalves [1], Frederico Augusto Pires Fernandes [3], Renato Baldan [4] and Artur Mariano de Sousa Malafaia [1]

[1] Campus Santo Antônio, São João del-Rei Federal University (UFSJ), Praça Frei Orlando, 170, Centro, São João del-Rei 36307-352, MG, Brazil

[2] Science and Technology of Minas Gerais, Federal Institute of Education, Ouro Branco, R. Afonso Sardinha, 90-Bairro Pioneiros, Ouro Branco 36420-000, MG, Brazil

[3] Center for Engineering, Modeling and Applied Social Sciences (CECS), Federal University of ABC (UFABC), Alameda da Universidade, s/n, São Bernardo do Campo 09606-045, SP, Brazil

[4] Campus of Itapeva, São Paulo State University (Unesp), Rua Geraldo Alckmin 519, Vila Nossa Senhora de Fátima, Itapeva 18409-010, SP, Brazil

[*] Correspondence: filipe.cunha2011@hotmail.com

**Abstract:** Nickel-based superalloys have excellent properties at high temperatures, which makes them appropriate for applications such as turbocharges and aeronautic gas turbines. MAR-M246 is a superalloy developed for these kinds of applications. The objective of this work was to study the total replacement of Ta by Nb in atomic percentage for this superalloy, regarding the oxidation resistance. Although both elements have the same role (form the $\gamma'$ precipitates) in nickel-based superalloys, Ta is more expensive than Nb. Thus, this replacement leads to the possibility of a price reduction. This work studied both conventional MAR-M246(Ta) and experimental MAR-M246(Nb) on cyclic oxidation tests at 900 and 1000 °C for up to 180 cycles. The oxide products were characterized by SEM/EDS and XRD analysis. The products of oxidation were mainly $TiO_2$, $Al_2O_3$, $Cr_2O_3$, $NiO$, and $Ni(Co)Cr_2O_4$. Mass gain variation per unit area was stable for both materials and temperatures. However, spalled areas were detected for tests performed at 1000 °C. The results obtained here suggest that use of Nb instead of Ta can be considered regarding high temperature oxidation for MAR-M246 superalloy.

**Keywords:** cyclic oxidation; MAR-M246; niobium; oxidation resistance; superalloys

## 1. Introduction

Nickel-based superalloys are known for their excellent properties at high temperatures. These materials have elevated yield strength, creep resistance, fatigue resistance, and also a great oxidation resistance, hence these superalloys are used in applications such as turbocharger engines and aero gas turbines [1–4]. All of these properties are directly related to the chemical composition of the alloy [5].

These alloys have a complex composition in which the alloying elements play a specific role. Elements such as Al, Ti, Ta, and Nb are responsible for the formation of ordered, stable $\gamma'$ precipitates that play a key role in the mechanical properties [3,6,7]. The presence of $\gamma'$ precipitates increases the creep resistance and yield strength as temperature rises, promoting the use of superalloys at high temperatures [5,8]. Furthermore, elements such as Al and Cr are related to the increase of oxidation resistance since they can form protective oxide layers [5,9]. Alloying elements also contribute to form carbides. If distributed along the grain boundary, these phases prevent grain–boundary sliding in polycrystalline Ni-based superalloys; hence, increasing mechanical resistance [7]. Elements such as Ti, Nb, Ta, Mo, W, Co, and Cr can act as carbide formers [7,10–12].

The conventional MAR-M246 alloy was created in the 1970's by Martin-Marietta corporation. This Ni-based superalloy is used in components such as automotive turbochargers, aerospace engines, diesel engines, and gasoline engines [12,13]. Due to chemical similarities, the replacement of Ta by Nb was recently proposed by our group, leading to the development of an experimental superalloy called Nb-modified MAR-M246 [3,6,7]. Alkmin et al. [13] studied both MAR-M246(Ta) and MAR-M246(Nb) for pseudo-isothermal oxidation tests at temperatures of 800, 900, and 1000 °C. For 800 and 900 °C, the experiments were performed for up to 1000 h, and for 1000 °C, they had a duration of 650 h. At 800 °C, these superalloys had very similar behavior, showing good oxidation resistance and parabolic-like curves, and also the same oxide products. For 900 °C, the conventional alloy exhibited a slightly higher oxidation resistance in comparison with the experimental alloy; moreover, MAR-M246(Nb) presented more inner oxidation, nitrides such as AlN, and a major Al depletion zone. For 1000 °C the MAR-M246(Ta) had a stable mass gain per unit area, while this was not observed in MAR-M246(Nb), which showed severe oxidation/spallation after 100 h of experiments. The authors also observed the formation of several oxides after the oxidation tests: $Cr_2O_3$, $TiO_2$, $Al_2O_3$, CoO, NiO, and $NiCr_2O_4$ spinels. Regarding other properties, the creep resistance for these two alloys was studied by Alkmin et al. [14]. They found out that the conventional superalloy has higher creep resistance compared to the experimental MAR-M246(Nb).

The literature is limited regarding the cyclic oxidation behavior of the MAR-M246 superalloy at high temperatures. In addition, the niobium-modified MAR-M246 developed in previous studies was tested only at quasi-isothermal oxidation conditions. The MAR-M246(Nb) is a newly developed superalloy designed by our research team. Therefore, papers regarding this material are limited and none of them studied it for cyclic conditions. Although both elements have the same function in nickel-based superalloys (form the $\gamma'$ precipitates), Ta is, generally, more expensive than Nb; thereby, the Nb-modified MAR-M246 may represent fabrication cost reduction [3,6,15,16].

The aim of this research is to compare the conventional and the experimental MAR-M246 regarding their cyclic oxidation behavior. As a secondary objective, it aims to study the oxidation kinetics and to characterize the oxidation products. The present study experimentally dealt with both MAR-M246(Ta) and MAR-M246(Nb) for cyclic oxidation tests at temperatures of 900 and 1000 °C. The mass variation curves were plotted and analyzed. Additionally, metallographic analysis, including energy-dispersive X-ray spectroscopy (EDS), X-ray diffraction (XRD), and scanning electron microscopy (SEM) of the top and cross-section of the alloys, was performed.

## 2. Materials and Methods

The MAR-M246(Nb) is an experimental superalloy and it was designed by replacing Ta with Nb atoms, which means that the number of tantalum atoms in MAR-M246(Ta) is the same as the number of niobium atoms in MAR-M246(Nb). The composition of the alloys in weight percentage is presented in the Table 1. The polycrystalline MAR-M246 samples were fabricated by vacuum induction melting (VIM) and the lost-wax technique using pre-heated ceramic molds (14 mm diameter and 130 mm length). Afterwards, to prepare the oxidation test samples, the superalloys were machined by wire electrical discharge machining in a disc shape of 8 mm diameter and 2 mm thickness.

**Table 1.** Chemical composition in wt. % of MAR-M246(Ta) and MAR-M246(Nb) superalloys.

| Elements (wt. %) | MAR-M246(Ta) | MAR-M246(Nb) |
|:---:|:---:|:---:|
| Cr | 9.0 | 9.09 |
| Co | 10.0 | 9.94 |
| Mo | 2.5 | 2.46 |
| W | 10.0 | 10.1 |
| Al | 5.5 | 5.44 |
| Ti | 1.5 | 1.44 |
| Ta | 1.5 | - |
| Nb | - | 0.84 |
| C | 0.15 | 0.15 |
| B | 0.015 | 0.01 |
| Zr | 0.05 | 0.05 |
| Ni | Bal. | Bal. |

Prior to the oxidation tests, the sample's surfaces were ground using SiC paper down to grit #600 and, subsequently, ultrasonically cleaned with isopropyl alcohol for 2 min; then the samples were dried using hot air; and, finally, their mass was measured using an analytical balance with a precision of $\pm 0.1$ mg. It is important to highlight that, due to the disk shape and small dimension of the samples, they were ground only at their flat faces.

The cyclic oxidation tests were performed in static air at 900 and 1000 °C for up to 180 cycles. Each cycle had 1 h at the test temperature and 10 min of cooling (air at room temperature < 30 °C). The tests were conducted in a vertical furnace with freedom to move up and down providing the required changes in the sample's temperature. Twelve samples were tested, six of each alloy. Six samples were used for the tests at 900 °C, three of each alloy. After 120 cycles, one sample was withdrawn, leaving two other specimens in the furnace. They were tested for 180 cycles, and, finally, one of them was characterized. Similarly, six samples were tested for 1000 °C, three of each material. By the end of 24 and 120 cycles, one specimen was withdrawn, leaving one sample inside the furnace. By the end of 180 cycles this sample was characterized. This paper focuses on showing the oxide layer characterization after 180 cycles of oxidation tests.

Mass measurements were conducted after a predefined number of cycles in order to obtain a curve of mass variation per unit area. Since only the lower and upper flat faces of the specimens were ground, it was noticed different oxidation behavior compared to the round surface. Thus, oxidation rates were not calculated and the mass variation curves were considered only for comparison between both alloys.

After the oxidation tests, selected specimens were prepared for oxide layer characterization. They were embedded in cold epoxy resin in order to protect the surface oxide layer and hot-mounted in bakelite resin. Afterwards, the specimens were ground using SiC paper down to grit #1200 and carefully polished with 1 μm and then 0.1 μm alumina suspension. Finally, the samples were ultrasonically cleaned with isopropyl alcohol and dried using hot air.

Thermodynamic simulations were carried out to predict the phase's stability and composition using JMatPro software (Ni-DATA version 7). Cross-section and topographic characterization analysis, including energy dispersive spectroscopy (EDS) and scanning electron microscopy (SEM), were carried out in the oxidized samples after 180 cycles. Not only was the aim of these analyses (SEM/EDS) to characterize the oxide layers, but also to characterize the carbides. The SEM images were performed using a backscattered electron mode detector, and both the images and the EDS analysis used 15 keV as the acceleration voltage. X-ray diffraction (XRD) was also conducted in order to determine the oxide's structure. This analysis was performed using a Cu-Kα radiation with an angular range of $10° < 2\theta < 100°$, using 0.02 step size and a scanning velocity of 2.0°/min.

## 3. Results

### 3.1. Mass Variation Kinetics

Figure 1 shows the mass variation per unit area for both superalloys and for both temperatures. Figure 1a presents the data for specimens exposed to cyclic oxidation tests at 900 °C for up to 180 cycles. This figure reveals that, after two cycles, the conventional MAR-M246(Ta) alloy lost mass while the modified MAR-M246(Nb) presented mass gain. Afterwards, both MAR-M246(Nb) and MAR-M246(Ta) had approximately the same mass gain per unit area between 2 and 10 cycles (0.309 and 0.324 mg/cm$^2$, respectively). Finally, for up to 180 cycles, the two alloys showed mass gain per unit area stabilization. This suggests that these materials show oxidation resistance in cyclic conditions at 900 °C. Alkmin et al. [13] observed similar results for up to 200 h of experiments at 900 °C. In their paper, the pair of materials had similar mass gain per unit area for up to 200 h (which is close to 180 cycles), although MAR-M246(Nb) gained a little bit more mass.

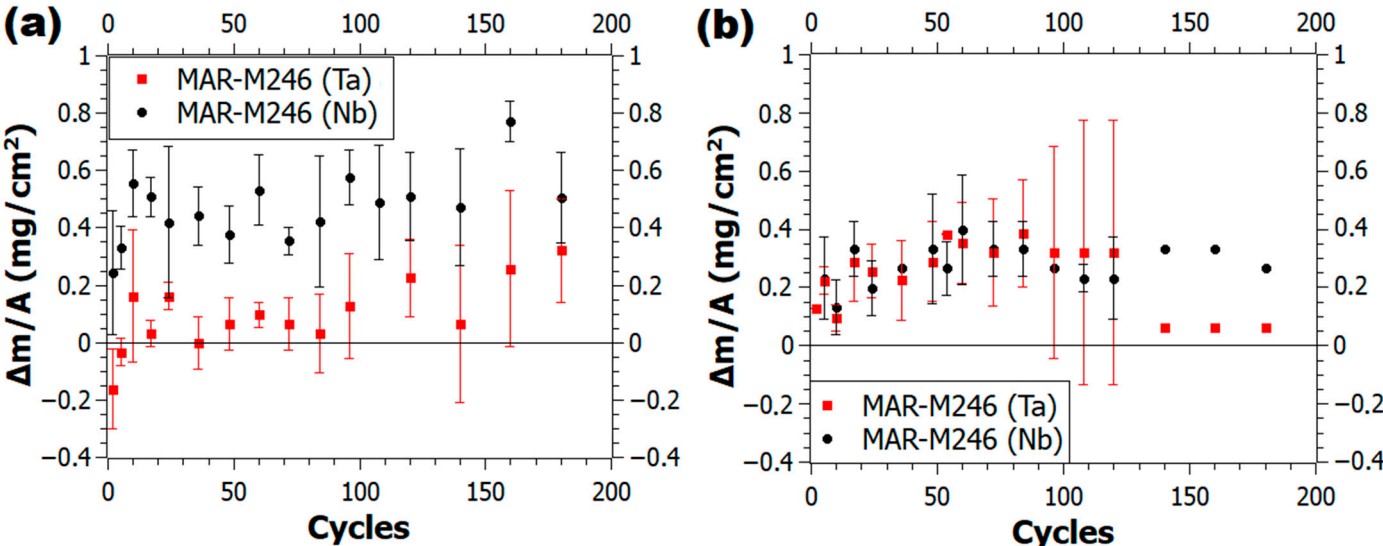

**Figure 1.** Comparison between the mass variation per unit area. (**a**) 900 °C, (**b**) 1000 °C.

Figure 1b presents the mass variation per unit area for samples exposed to cyclic oxidation tests at 1000 °C for up to 180 cycles. In comparison to the experiments at 900 °C, the materials showed an even closer behavior. Figure 1b indicates a lot of intersections between the standard deviation for these superalloys until cycle 120. After this cycle, a specimen of each alloy was collected for characterization procedures. MAR-M246(Ta) presented higher standard deviations for the 96, 108, and 120 cycles. This suggests that one of the samples suffered mass loss during the experiments. Although mass loss may occur during the process, these two superalloys showed mass gain stabilization; therefore, both materials exhibit good oxidation resistance for cyclic conditions at 1000 °C. On the other hand, Alkmin et al. [13] obtained different results at 1000 °C, whereby the conventional MAR-M246 also exhibited stabilization of mass gain, while the Nb-modified MAR-M246 presented a higher mass gain up to 200 h, followed by continuous mass loss, due to severe oxide layer spallation. Thus, the results presented here may indicate that, in cyclic conditions, the MAR-M246(Nb) shows higher oxidation resistance than in pseudo-isothermal conditions, at least for up to 200 h (or approximately 180 cycles). Alkmin et al. [13] showed that MAR-M246(Nb) had a severe mass loss after 200 h, it may be interesting to study long-term cyclic oxidation behavior (more than 180 cycles) to better understand the Nb-modified superalloy under cyclic conditions and verify whether this alloy does not exhibit mass loss after 180 cycles.

The MAR-M246(Nb) (0.84 wt% of Nb) and MAR-M246(Ta) (1.5 wt% of Ta) exhibited a stable mass variation per unit area; thus, these materials had a good oxidation resistance,

which may be related to the Nb and Ta content. In general, a small wt% of Nb (less than 2% in wt%) and an increasing wt% of Ta can improve the oxidation resistance. Mallikarjuna et al. [17] studied Ni-based superalloys exposed to an isothermal oxidation test at 900 °C. Two of those alloys had 1.7 and 6.4 wt% of Ta. Although both showed oxidation resistance and parabolic behavior, the one with higher Ta content exhibited less mass variation per unit area. Smialek and Bonacuse [18] also observed something similar by studying several Ni-based superalloys exposed to cyclic oxidation testes at 1100 °C. They indicated that, by increasing the wt% of Ta, the oxidation resistance was enhanced. Additionally, these authors showed that by increasing the wt.% of Nb, the mass loss per unit area was higher. Weng et al. [19] compared three Ni-based superalloy which were exposed to oxidation tests at 800 °C. The only difference between these alloys was the wt% of Nb: 0.0, 2.0, and 2.5%. The alloy that contained 2.0 wt% of Nb showed the highest oxidation resistance. Analyzing this data from the literature, it can be suggested that the low Ta value present in the conventional MAR-M246 does not promote high-temperature oxidation, while low Nb content could improve high-temperature oxidation. Furthermore, no considerable difference was observed in the mass variation curves.

### 3.2. Topographic and Cross-Section SEM/EDS Analysis

The typical microstructures for both superalloys are similar: carbides are usually found in the interdentritic region, the distance between the dendritic areas are similar, and eutectic pools can be observed, as pointed out by Alkmin et al. [13]. Since the typical microstructures were presented by our team in previous work [13], this paper does not aim to deeply discuss this topic, giving more attention to oxide products. In order to gain knowledge of the behavior of the conventional and Nb-modified superalloys, SEM analysis was conducted after cyclic oxidation tests. The analyses were conducted after 180 cycles on the specimen's surface and cross-section for the two alloys at 900 and 1000 °C. The topographic SEM/EDS analysis for the two superalloys after 180 cycles of oxidation at 900 °C is shown at Figure 2. Both MAR-M246(Ta) and MAR-246(Nb) exhibit similar characteristics. Region 1 of Figure 2a,b is mostly constituted by Al and O, possibly an Al-rich oxide. This region is also characterized by a dark grey color. Region 2 of Figure 2a,b has a totally different morphology in comparison with region 1. This region depicts randomly distributed oxide islands grown across the specimen's surface. EDS mapping indicates that region 2 is primarily constituted by O, Cr, and Ti.

Figure 3 shows the cross-section SEM/EDS analysis for the two superalloys after 180 cycles at 900 °C. Both superalloys had a formation of two well-defined oxide layers: outer and inner. The outer oxide layer (for both Figure 3a,b) is mostly constituted by O, Cr, and Ti. This oxide layer was also observed in Figure 2 in region 2. The inner oxide layer is mainly constituted by Al and O, possible an Al-rich oxide. The formation of $Al_2O_3$ and $Cr_2O_3$ can be facilitated by the presence of Ta and Nb. It can explain the dense Al-rich and Cr-rich oxides observed [17,20,21]. Figure 3a indicates a small carbide, and Ti is present in this structure, as also highlighted by Baldan et al. [12]. Figure 3b shows other carbides in the MAR-M246(Nb) superalloy. These carbides are mainly constituted by Ti and Nb, according to the EDS maps. Tantalum usually acts like a carbide former. Once Nb and Ta have similar functions [12,22], it is reasonable to consider that Nb is also a carbide-former. Alkmin et al. [14] also found out that Nb is an important element in carbide formation. Different Ni-based superalloys also presented Nb as a carbide-former, as reported elsewhere [10,11,23].

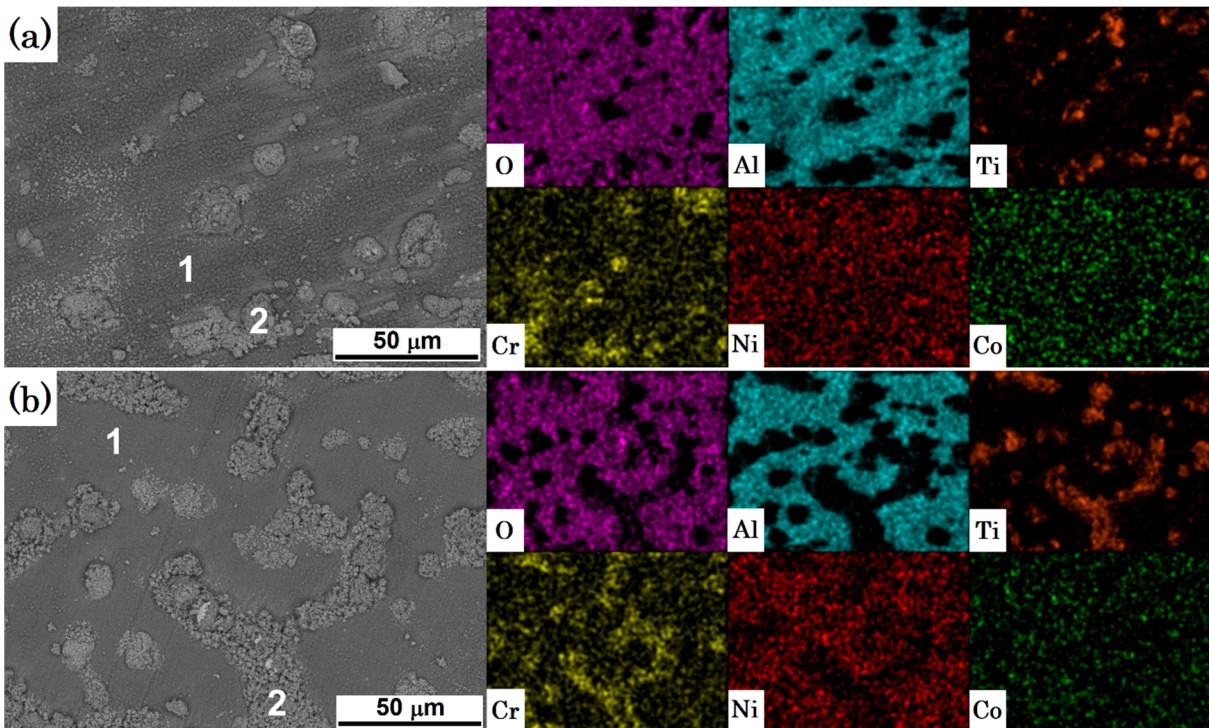

**Figure 2.** SEM and EDS mapping of elements for 180 cycles at 900 °C on the top of the specimens. (**a**) MAR-M246(Ta), (**b**) MAR-M246(Nb).

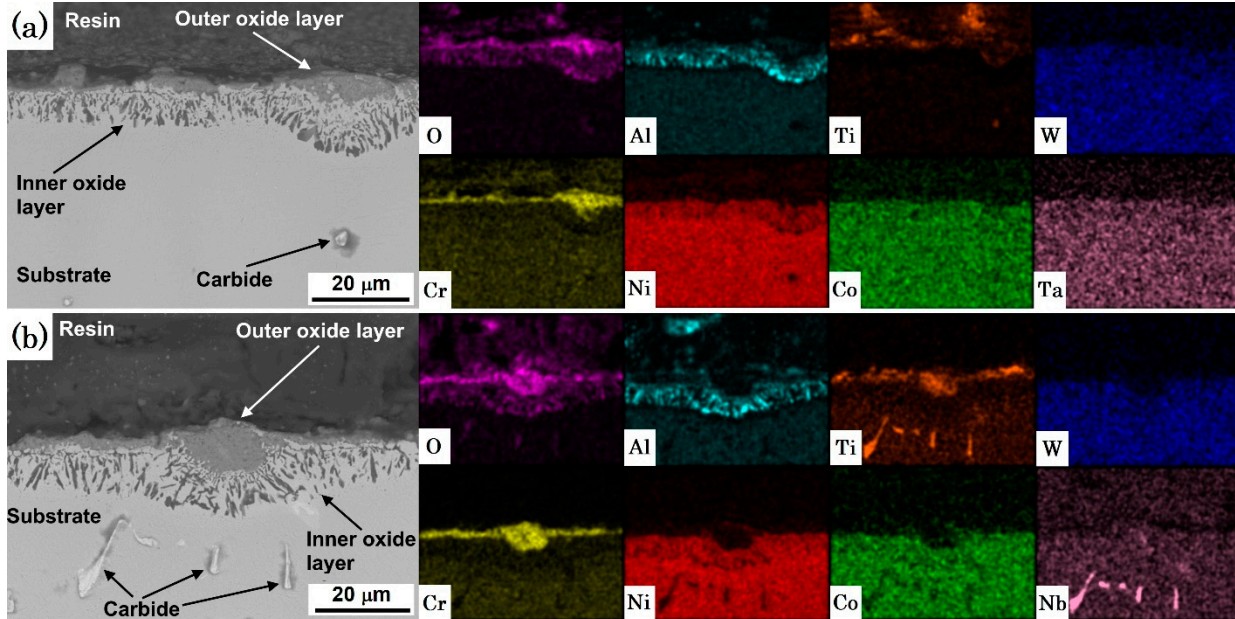

**Figure 3.** SEM and EDS mapping of elements for 180 cycles at 900 °C on the cross-section of the specimens. (**a**) MAR-M246(Ta), (**b**) MAR-M246(Nb).

It is important to highlight that the superalloy had similar aspects for both top and cross-section analysis at 900 °C for up to 180 cycles. Even after replacing Ta with Nb, the two alloys maintained similar oxide composition and oxide morphology. This indicates that the replacement did not interfere with these aspects of the superalloy. However, the carbide composition for these two superalloys showed different characteristics.

The top SEM/EDS analysis for the two superalloys for up to 180 cycles at 1000 °C is shown in Figure 4. These analyses are more complex than the analysis shown previously

for 900 °C. This indicates the influence of the change in temperature. Both Figure 4a,b show similar characteristics. Region 1 was constituted of O, Al, and Cr. Region 2 had a higher intensity of Cr and Ti than region 1, although this also indicates the presence of elements such as Ni, Co, and O. Region 3 seemed to be mostly constituted by W and O, a possible W-rich oxide. This oxide was probably a $W_{20}O_{58}$, which usually evaporates in the form of $WO_{3(g)}$. This evaporation process can reduce the oxide layer stability [22,24], therefore, explaining in part, the presence of spalled areas in region 5, as will be soon discussed. Region 4 was mostly constituted by Al and O, possibly an Al-rich oxide such as $Al_2O_3$. This region is similar to region 1 in Figure 2 because of its high content of Al and dark grey color. Finally, region 5 is probably a spalled area. It can be justified by the high content of the primary elements that constitute the alloys (Co, W, and Ni, as can be seen in Table 1) and also by the absence of O. Similar results were found in previous papers [13,25]. Baldan et al. [25] also observed spalled areas for the conventional MAR-M246 exposed for isothermal oxidation tests at 1000 °C for 240 h. Additionally, Alkmin et al. [13] found spalled areas for MAR-M246(Nb) pseudo-isothermally tested at 1000 °C for 650 h.

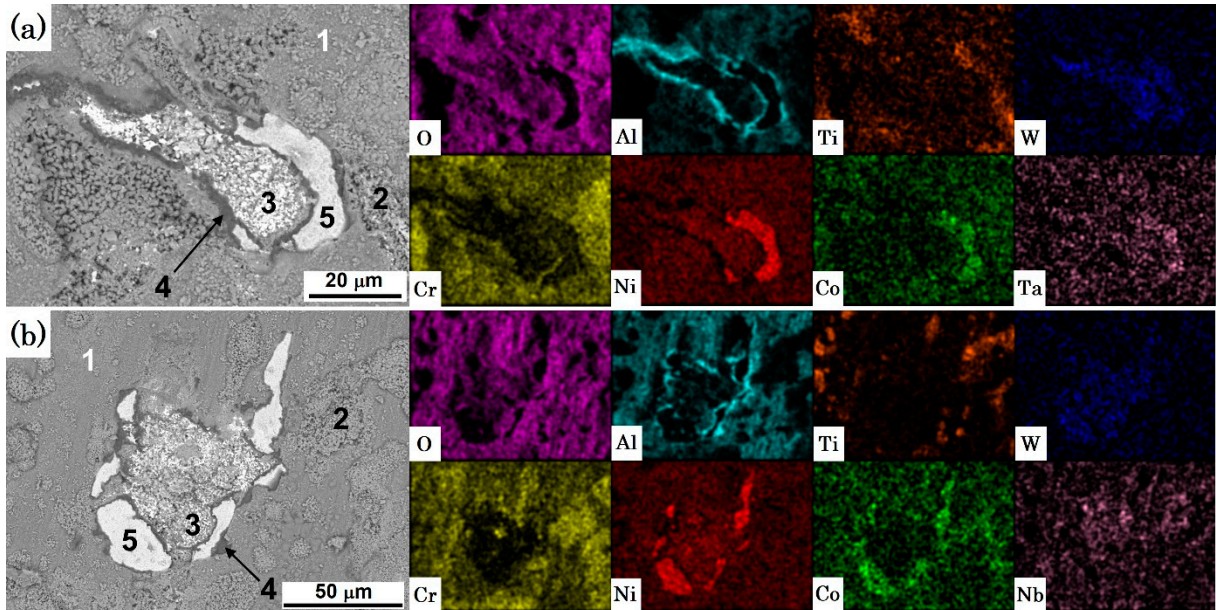

**Figure 4.** SEM and EDS mapping of elements for 180 cycles at 1000 °C on the top of the specimens. (**a**) MAR-M246(Ta), (**b**) MAR-M246(Nb).

Figure 5 shows the cross-section SEM/EDS analysis for the two superalloys oxidized for up to 180 cycles at 1000 °C. Both MAR-M246(Ta) and MAR-M246(Nb) had a formation of two well-defined outer and inner oxide layers. Two different oxide layers were also observed for the tests conducted at 900 °C; however, there is an evident difference in the layers' morphology. The outer oxide layer (for both Figure 5a,b) is mostly constituted by O, Co, Cr, Ti, and probably Ni. Possibly, this layer represents region 2, which was analyzed in Figure 4. The inner oxide layer (for both Figure 5a,b) is mainly constituted by Al-rich oxides, probably $Al_2O_3$. This is the layer immediately over the substrate, and it is related to region 4, as shown in Figure 4. Figure 5a (MAR-M246(Ta)) indicates two interesting structures: regions 1 and 2. The arrows show region 1, which is a Ti-rich area. This region is probably a titanium nitride. Although nitrogen was not identified by EDS analysis, basically due to technical limitation, very similar morphologies were identified in previous studies [13,25], which also studied MAR-M246 superalloy. The region indicated by number 2 is constituted by the most present elements in the superalloy (Ni, Co, and W) and O is not observed; therefore, it is part of the substrate surrounded by oxides. This phenomenon is known as incorporation of metal islands into the oxide layer. This process is well described by Das et al. [26].

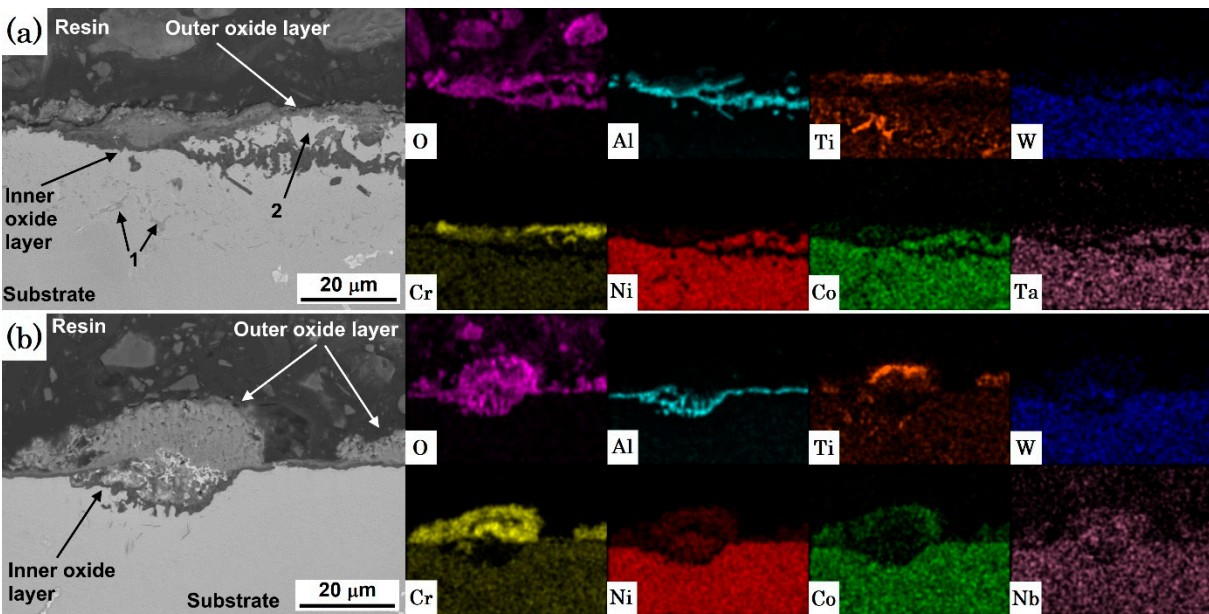

**Figure 5.** SEM and EDS mapping of elements for 180 cycles at 1000 °C on the cross-section of the specimens. (**a**) MAR-M246(Ta), (**b**) MAR-M246(Nb).

*3.3. XRD Analysis*

Figure 6 reveals the XRD analysis for both MAR-M246(Ta) and MAR-M246(Nb), tested by cyclic oxidation at 900 and 1000 °C up to 180 cycles. The formed oxides are alike for both superalloys and temperatures. The only difference is the absence of the $Ni(Co)Al_2O_4$ spinel for the MAR-M246(Ta) superalloy at 900 °C. Table 2 shows all the oxides and spinels identified by the XRD analysis. Figure 6a,b show the oxide layer evolution for the conventional MAR-M246 for 900 and 1000 °C, respectively. Figure 6a indicates that the most intense peaks (approximately at 44°, 51°, 75°, and 92°) are related to Ni. This suggests that the oxide layer was thin because of the detection of the substrate, mostly constituted by this element. Figure 6b reveals that increasing the temperature led to the development of $TiO_2$, $Cr_2O_3$, $Al_2O_3$, and $Ni(Co)Cr_2O_4$ phases. This can be concluded by analyzing the peaks at approximately 27°, 30°, 36°, and 58°. These peaks are more intense at 1000 °C than at 900 °C. Moreover, at 1000 °C, the peak at 36° (related to $Ni(Co)Cr_2O_4$) is the most intense peak; in other words, it represents the increase in the oxide layer's complexity.

Figure 6c,d show the oxide layer evolution for the experimental MAR-M246 for 900 and 1000 °C, respectively. At 900 °C, Figure 6c, the MAR-M246(Nb) shows similar behavior in comparison to the conventional alloy at this same temperature. The most intense peak (around 44°) is related to Ni. This suggests that the oxide layer was thin, due to Ni detection. Despite this similarity, the Nb-modified superalloy exhibited more intense peaks (approximately at 24°, 27°, 34°, 36°, and 42°) for $TiO_2$ and $Cr_2O_3$, in comparison to MAR-M246(Ta) at 900 °C. This suggests that replacing Ta with Nb facilitates the formation of these oxides. Figure 6d shows the X-ray diffraction analysis for the MAR-M246(Nb) at 1000 °C. The increase in temperature caused a similar impact for both MAR-M246(Ta) and MAR-M246(Nb). At 1000 °C, the most intense peak is related to $Ni(Co)Cr_2O_4$ and $Cr_2O_3$ at approximately 36°. This spinel shows more intense peaks (54° and 58°) at 1000 °C than at 900 °C for the experimental superalloy. Finally, replacing Ta with Nb did not lead to considerable changes in the oxides formed, as observed by the DRX analysis at 1000 °C, because Figure 6b,d are alike.

**Table 2.** Oxidation products obtained by XRD diffractograms for 180 cycles.

| Alloy | Cycles | Oxidation Products at 900 °C | Oxidation Products at 1000 °C |
|---|---|---|---|
| MAR-M246(Ta) | 180 | $TiO_2$, $Al_2O_3$, $Cr_2O_3$, NiO, Ni(Co)$Cr_2O_4$ | $TiO_2$, $Al_2O_3$, $Cr_2O_3$, NiO, Ni(Co)$Cr_2O_4$, Ni(Co)$Al_2O_4$ |
| MAR-M246(Nb) | 180 | $TiO_2$, $Al_2O_3$, $Cr_2O_3$, NiO, Ni(Co)$Cr_2O_4$, Ni(Al)$Cr_2O_4$ | $TiO_2$, $Al_2O_3$, $Cr_2O_3$, NiO, Ni(Co)$Cr_2O_4$, Ni(Co)$Al_2O_4$ |

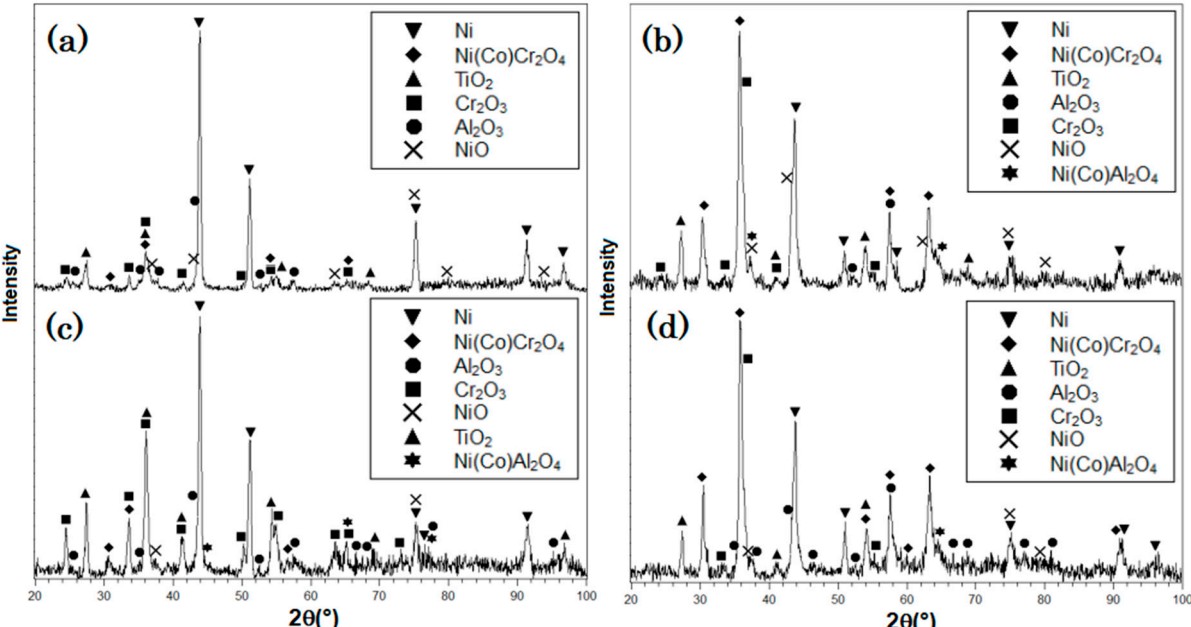

**Figure 6.** XRD diffractograms for 180 cycles. (**a**) MAR-M246(Ta) at 900 °C, (**b**) MAR-M246(Ta) at 1000 °C, (**c**) MAR-M246(Nb) at 900 °C, and (**d**) MAR-M246(Nb) at 1000 °C.

*3.4. Thermodynamic Simulations*

In addition to oxidation behavior, the Ta replaced by Nb can affect the phase formation in the substrate. Thus, thermodynamic simulation was performed by JMatPro, and Figure 7 shows the phase stability between 600 and 1400 °C thermodynamic simulation. As can be seen in Figure 7, the expected phases are the same and in similar amounts for both MAR-M246(Ta) and MAR-M246(Nb). Similar results were also obtained by Alkmin et al. [14], who observed similar phases for both alloys using ThermoCalc software for simulation. They also performed a Scheil simulation to evaluate solidification and observed that, in addition to γ and γ′, only boride and MC carbide were expected to form. On the other hand, as observed here, regarding the oxidation test temperatures, the primary carbide, MC, is not stable at 900 and 1000 °C, while $M_6C$ and $M_{23}C_6$ are. For the pair of superalloys, the γ′ is mainly formed by Al. Although Ta and Nb are γ′ precipitate formers, they contribute only 0.6 to 1% (at%) of the composition of the phase.

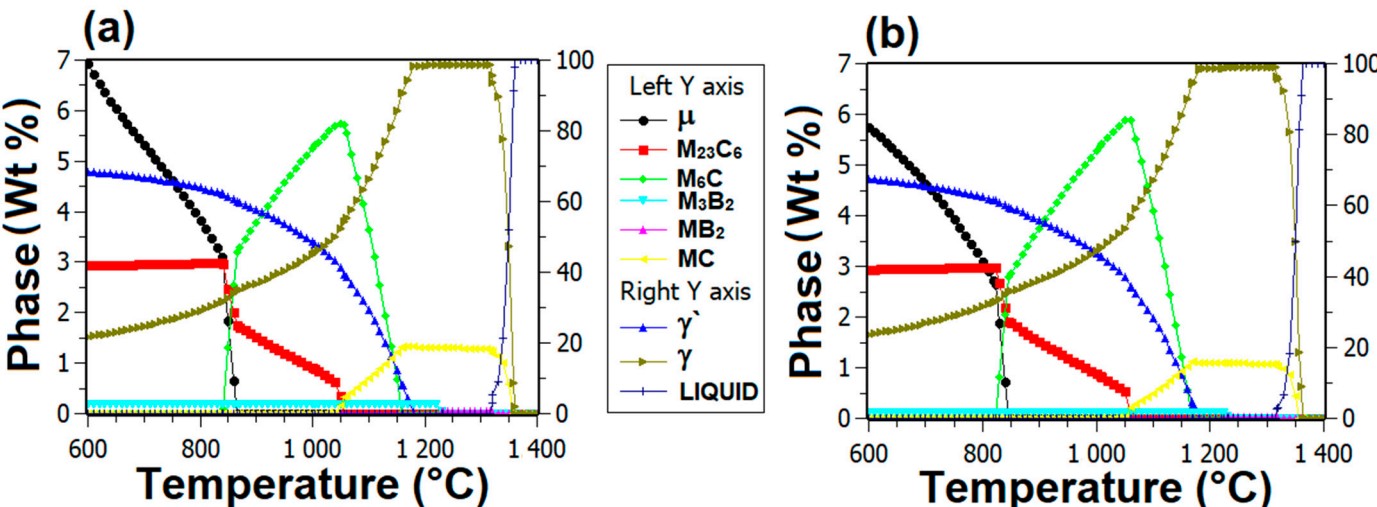

**Figure 7.** Thermodynamic simulations for phase variation. (**a**) MAR-M246(Ta), (**b**) MAR-M246(Nb).

To evaluate the presence of Ta and Nb in carbides, thermodynamic simulation and SEM/EDS analysis were used. According to the simulations for carbide composition (shown on Figure 8), Nb and Ta were elements present in the primary MC carbides. These structures are not stable at 900 and 1000 °C, as observed in Figure 7; therefore, carbides constituted by these elements must have been formed during the solidification process, as mentioned before. Figure 8a,b show the composition for MC carbides for both MAR-M246(Ta) and MAR-M246(Nb), respectively. These images suggest that the MC carbide is mainly formed by Ta, Nb, Ti, and W. This result converges with the information exhibited in previous studies [14,25]. Comparing these two superalloys, it can be observed that the at% of Ta and Nb in MC carbides are similar in the respective alloys. For the other phases ($M_6C$ and $M_{23}C_6$), tantalum and niobium do not play a significant role for temperatures of 900 and 1000 °C.

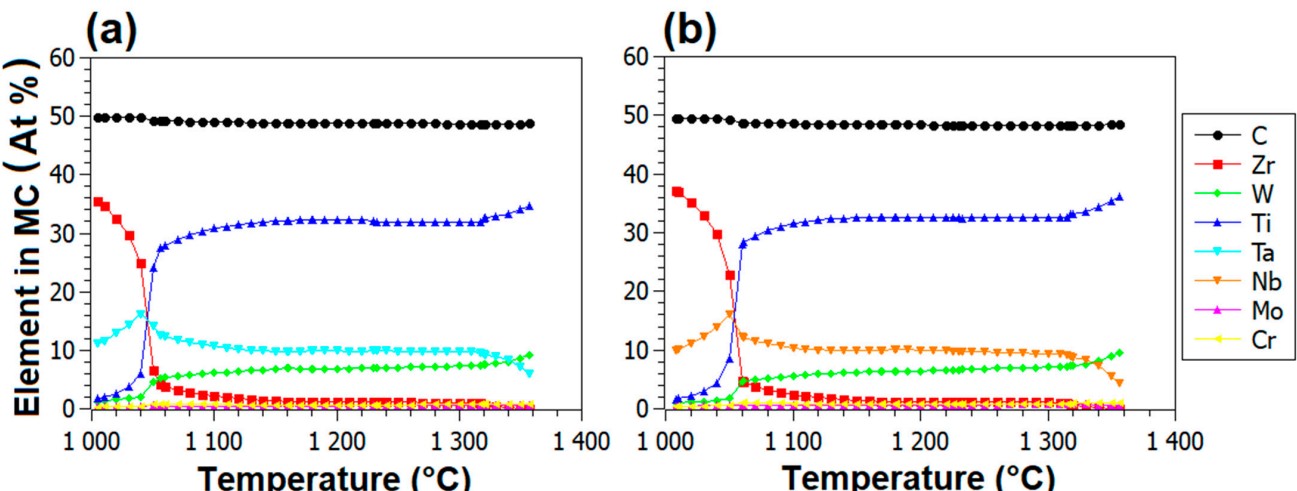

**Figure 8.** Thermodynamic simulations for carbide composition. (**a**) MAR-M246(Ta), (**b**) MAR-M246(Nb).

Figure 9 focuses on showing the differences between the carbides, possibly MC, formed on both superalloys. These analyses show carbides that are observed for specimens exposed to cyclic oxidation tests performed for up to 120 cycles at 900 and 1000 °C. First, it is important to remember that the experimental MAR-M246(Nb) was designed by replacing the number of atoms of Ta by Nb. It is evident the difference in the intensity of Ta and Nb in the formed carbides. It indicates that there are more niobium atoms constituting the

carbides than in solid solution in the MAR-M246(Nb) alloy. In order to better understand this phenomenon, regions 1 and 2 (Figure 9a,b) were analyzed by EDS. Regions 1 and 2 in Figure 9a indicate 8.1 and 6.8% of Ta in at%, respectively, while in Figure 9b, the results for regions 1 and 2 are 20.8 and 19.3% of Nb in at%, respectively. Although these values may not describe exactly the real amount of those elements, these atomic percentages are useful for comparing the two superalloys and their carbides. These results diverge from the thermodynamic calculations (Figure 8). Since the simulations consider a perfect equilibrium through the solidification process, and MC carbides are not stable at 900 and 1000 °C (that is the case for Figure 9), some differences between calculations and SEM/EDS are expected. Alkmin et al. [14] also determined that niobium is more present in at% in carbides than tantalum. This difference in carbide formation, however, did not appear to affect oxidation behavior. Future studies should be conducted to evaluate in detail the microstructural aspects of gamma and gamma prime fractions, as Ta and Nb were also present in these main phases.

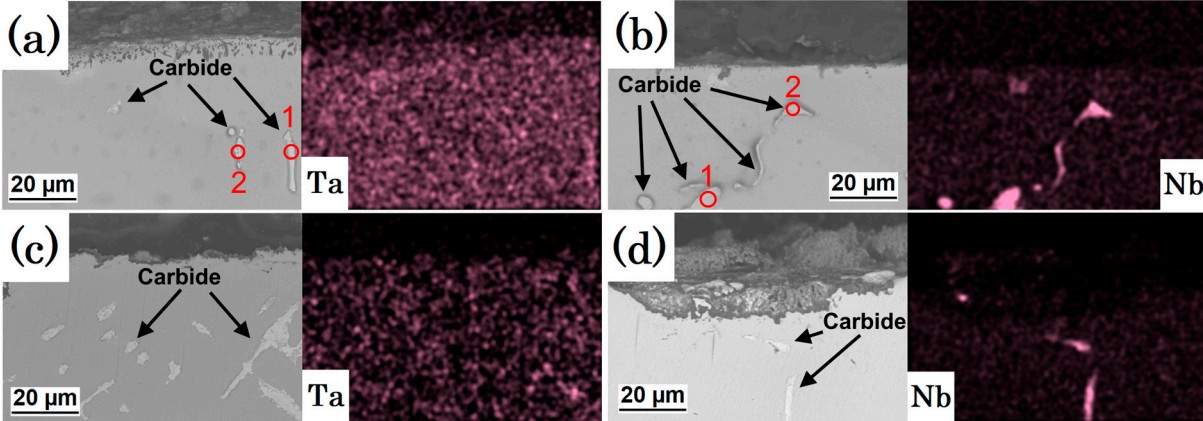

**Figure 9.** SEM and EDS mapping for 120 cycles. (**a**) MAR-M246(Ta) at 900 °C, (**b**) MAR-M246(Nb) at 900 °C, (**c**) MAR-M246(Ta) at 1000 °C, and (**d**) MAR-M246(Nb) at 1000 °C.

## 4. Conclusions

The cyclic oxidation behavior of two Ni-based superalloys at 900 and 1000 °C for up to 180 cycles was studied. The aim of this study was to compare the performance of a conventional MAR-M246 and an experimental version of this superalloy, called niobium-modified MAR-M246, in which Ta atoms are replaced with Nb ones. The following conclusions can be drawn.

Around the spalled areas, which were observed on the surface of both superalloys for cyclic oxidation tests at 1000 °C, W-rich oxides were detected. This may indicate that the formation of this oxide or even its evaporation can induce spalled areas. Therefore, W oxides seem to be detrimental for the oxide layer stabilization regarding the two studied alloys.

In both temperatures the two superalloys showed mass stabilization and the oxidation products presented similar morphology and composition when evaluated topographically and cross-sectionally. At 900 °C, no spalled area was observed and both superalloys exhibited good oxidation resistance. The oxide layer was constituted by an internal layer of $Al_2O_3$. Moreover, an external oxide layer was also observed, which was Cr- and Ti-rich, indicating the existence of $TiO_2$ and $Cr_2O_3$. Finally, NiO and spinel formation were also observed. At 1000 °C, despite some spalled areas, the mass variation per unit area also indicated oxidation resistance up to 180 cycles. The oxide layer was formed by an inner oxide layer of $Al_2O_3$ and an outer oxide layer, containing $TiO_2$, $Cr_2O_3$, NiO, and $Ni(Co)Cr_2O_4$. Similar results were observed at 900 °C. However, despite these similarities, the increase in temperature induced marked morphology changes, such as spalled areas,

which was already mentioned, and W-rich oxides. Further studies with longer periods of cyclic oxidation tests could be useful for evaluating the long-term performance.

Although these superalloys exhibit a lot of similarities, indicating that replacing Ta with Nb did not lead to significant changes in the MAR-M246 superalloy, there are some differences between these two superalloys that must be pointed out. First, the formation of $TiO_2$ and $Cr_2O_3$ seems to be facilitated in MAR-M246(Nb) at 1000 °C, in comparison to the conventional MAR-M246. Second, there are more Nb in at% in the carbides of the experimental alloy than Ta in at% in the carbides of the conventional alloy. The impact of this higher content of Nb in carbides should be further investigated. Finally, the results presented here suggest that Ta can be replaced by Nb in MAR-M246 superalloy, considering cyclic oxidation performance.

**Author Contributions:** Conceptualization, R.B. and A.M.d.S.M.; methodology, R.B. and A.M.d.S.M.; software, R.B.; formal analysis, F.A.d.F.C., R.d.A.R., and S.P.G.; investigation, R.d.A.R. and S.P.G.; resources, R.B. and A.M.d.S.M.; data curation, F.A.d.F.C., R.d.A.R. and S.P.G.; writing—original draft preparation, F.A.d.F.C.; writing—review and editing, F.A.P.F., R.B., and A.M.d.S.M.; visualization, F.A.d.F.C. and R.d.A.R.; supervision, F.A.P.F., R.B. and A.M.d.S.M.; project administration, R.B. and A.M.d.S.M.; funding acquisition, R.B. All authors have read and agreed to the published version of the manuscript.

**Funding:** São Paulo Research Foundation (FAPESP) for the project—grant no. 2018/07802-9—project resources. Coordination for the Improvement of Higher Education Personnel (CAPES)—grant no. 001.

**Conflicts of Interest:** The authors declare no conflict of interest.

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
