# Peer review of "Cyclic Oxidation Behavior of Conventional and Niobium-Modified MAR-M246 Superalloy at 900 and 1000 °C"

_coatings, doi:10.3390/coatings13030519_

Round 1

Reviewer 1 Report

The objective of this work was to study the total replacement of Ta by Nb in atomic percentage for this superalloy regarding the oxidation resistance. This work studied both conventional MAR-M246(Ta) and experimental MAR-M246(Nb) on cyclic oxidation tests at 900 and 1000 °C for up to 180 cycles. Mass gain variation per unit area was stable for both materials and temperatures. The results obtained here suggest that use of Nb instead of Ta can be considered regarding high temperature oxidation for MAR-M246 superalloy. Conclusions are well established and agree with analysis results and their discussion. The experiment results are interesting but the following points are to be considered for modification:

1) The objective of this work was to study the total replacement of Ta by Nb in atomic percentage for this superalloy regarding the oxidation resistance. However, in addition to oxidation resistance, whether other properties to meet the requirements? Such as yield strength, creep resistance, fatigue resistance.

2) In Figure 5, the depth of outer oxide layer are different, whether it can be considered that use of Nb instead of Ta for MAR-M246 superalloy.

3) Whether the γ’ precipitates are different, if Ta was instead of Nb for MAR-M246 superalloy.

4) The results need to be better connected to the available literature, providing some comparison with published studies.

5) Please better highlight the innovative aspects of the current work.

Reviewer 2 Report

The manuscript's subject is generally interesting, and the text is well organized into paragraphs. The results give important information for future investigations and applications. The introduction provides relevant background information and sufficiently presents the research motivation. The results and conclusions are presented concisely and adequately demonstrate the achievement of the research objective. However, some minor corrections are needed to improve the manuscript's clarity.

·         In the Introduction chapter (line 44), the Authors indicate the positive aspects of carbide formation, preventing grain-boundary sliding. This is only in the case of superalloys in polycrystalline form; in monocrystalline superalloys, carbides are undesirable. It would be good to mention it and indicate at the start that the tests were carried out on polycrystalline samples of superalloys.

·         At the start of the Materials and Methods chapter (lines 78,79), there is an unnecessary repetition of the aim of the experiments. Please move it to the previous chapter and match the repeated sentences.

·         Please add a few sentences about the casting production method in the Materials and Methods section. Was it slow cooling with the furnace to obtain polycrystalline superalloy cylinders or another way?

·         Figures 2, 3, 4, and 5 (captions) – please correct the sentence; it should be: top of the specimen, not the superalloy.

·         line 270 – please explain the DRX abbreviation.

·         line 289 – please remove the comma in the temperature notation.

Reviewer 3 Report

In the present research, the authors try to investigate the cyclic oxidation behavior of MAR-M246 superalloy with and without niobium modification at 900 and 1000 °C. The paper exhibits some results but there are some questions in the present state.

1. The title is some confusing. The title of “Cyclic oxidation behavior of conventional and niobium-modified MAR-M246 superalloy at 900 and 1000 °C” may be much better.

2. In the experimental, the authors are suggested to introduce the smelting of the niobium-modified MAR-M246 superalloy.

3. In the content, the authors are suggested to give the typical microstructure of the superalloy with and without niobium modification, because the Ta and Nb is the main elements of carbides. As the authors’ research, the carbides could influence the oxidation.

4. In the Figure 7, the words “wt% Phase” in the y-coordinate are suggested to change into “ Phase (wt. %)”. The dot in the “1.000” could be deleted.

5. In the Figure 8, the words “At % Element in MC” in the y-coordinate are suggested to change into “ Element in MC (at. %)”. The dot in the “1.000” could be deleted.

6. In the figure caption of Figure 1, the curves is not appropriate.

7. The SEM observation on the surface oxides with high magnification could be given for better understanding.

8. The description of “The region 1 of Figure 2a and 2b is mostly constituted by Al and O,  possibly an Al-rich oxide like Al2O3.” is not appropriate. The authors should give the detailed EDS analysis on the detailed region to support the opinion.

9. Based on the authors’ observations on 120 cycles oxidation, the MAR-M246(Ta) at 900 °C has obvious inner oxidation, but the MAR-M246(Nb) at 1000 °C has the obvious inner oxidation. It wonders why the alloying element replacing have induced such a oxidation behavior? The authors are suggested to give more discussion on the phenomenon.    

10. The conclusion is some simply. More opinions should be concluded based on the results.

11. In the content, the authors are suggested to give the schematic to describe the cycle oxidation behavior of the Nb-modified MAR-M246 superalloy, which would improve the manuscript obviously. 

Reviewer 4 Report

The paper presents the results of research on two nickel alloys: the standard MAR-M246 and the experimental one containing niobium instead of tantalum. The susceptibility to cyclic oxidation of both alloys was evaluated by measuring the mass loss, as well as evaluating the surface features using the EDS method. Oxidation products were identified by XRD. The type of carbides formed in the microstructure of the tested alloys was modeled, taking into account the different contents of alloying elements. No significant differences were observed between the tested alloys.

The presented research results should be considered as preliminary. From the point of view of high-temperature operation of both alloys, it would be important to determine stability at high temperatures, such as creep resistance or fatigue strength. Nevertheless, I consider the work as well prepared and worth publishing.

Minor comment:

1) the description of the SEM research requires the details of the experiment to be indicated: detector, accelerating voltage

Round 2

Reviewer 1 Report

The paper has been revised, it can be accepted now. 

Author Response

Dear, Reviewer.

Thank you for your suggestions 

Kind regards

Reviewer 3 Report

The authors have revised the manuscript according to the suggestions and answered the questions. Now, it is improved and could be accepted.

Author Response

(The authors gave the same response as above.)
